# Discriminative Particle Filter Reinforcement Learning for Complex Partial Observations

**Xiao Ma**[1]**, Peter Karkus**[1]**, David Hsu**[1] **and Wee Sun Lee**[1]**, Nan Ye**[2]
[1]National Unviersity of Singapore, [2]The University of Queesland
{xiao-ma,karkus,dyhsu,leews}@comp.nus.edu.sg, nan.ye@uq.edu.au

## Abstract

Deep reinforcement learning is successful in decision making for sophisticated games, such as Atari, Go, etc. However, real-world decision making often requires reasoning with partial information extracted from complex visual observations. This paper presents *Discriminative Particle Filter Reinforcement Learning* (DPFRL), a new reinforcement learning framework for complex partial observations. DPFRL encodes a differentiable particle filter in the neural network policy for explicit reasoning with partial observations over time. The particle filter maintains a belief using learned discriminative update, which is trained end-to-end for decision making. We show that using the discriminative update instead of standard generative models results in significantly improved performance, especially for tasks with complex visual observations, because they circumvent the difficulty of modeling complex observations that are irrelevant to decision making. In addition, to extract features from the particle belief, we propose a new type of belief feature based on the moment generating function. DPFRL outperforms state-of-the-art POMDP RL models in Flickering Atari Games, an existing POMDP RL benchmark, and in *Natural* Flickering Atari Games, a new, more challenging POMDP RL benchmark introduced in this paper. Further, DPFRL performs well for visual navigation with real-world data in the Habitat environment. The code is available online [1].

## 1 Introduction

Deep Reinforcement Learning (DRL) has attracted significant interest, with applications ranging from game playing (Mnih et al., 2013; Silver et al., 2017) to robot control and visual navigation (Levine et al., 2016; Kahn et al., 2018; Savva et al., 2019). However, natural real-world environments remain challenging for current DRL methods (Arulkumaran et al., 2017), in part because they require (i) reasoning in a partially observable environment and (ii) reasoning with complex observations, such as visually rich images. Consider, for example, a robot, navigating in an indoor environment, with a camera for visual perception. To determine its own location and a traversable path, the robot must extract from image pixels relevant geometric features, which often coexist with irrelevant visual features, such as wall textures, shadows, etc. Further, the task is partially observable: a single image at the current time does not provide sufficient features for localization, and the robot must integrate information from the history of visual inputs received.

The *partially observable Markov decision process* (POMDP) provides a principled general framework for decision making under partial observability. Solving POMDPs requires tracking a sufficient statistic of the action-observation history, e.g., the posterior distribution of the states, called the *belief*. Most POMDP reinforcement learning (RL) methods summarize the history into a vector using a recurrent neural network (RNN) (Hausknecht & Stone, 2015; Zhu et al., 2018). RNNs are model-free generic function approximators. Without appropriate structural priors, they need large amounts of training data to learn to track a complex belief well.

Model-based DRL methods aim to reduce the sample complexity by learning a model together with a policy. In particular, to deal with partial observability, Igl et al. (2018) recently proposed DVRL, which learns a generative observation model embedded into the policy through a Bayes filter. Since

---
[1]https://github.com/Yusufma03/DPFRL

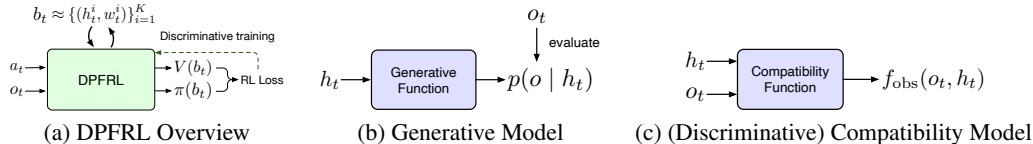

Figure 1: (a) DPFRL tracks a learned latent belief with a differentiable particle filter and learns a policy conditioned on the particle belief. (b) Generative models learn $p(o \mid h_t)$, the distribution of observations $o$ given the latent state $h_t$. The particle filter then evaluates the probability of an observed $o_t$ to get a compatibility measure of $o_t$ and $h_t$. (c) The compatibility function directly predicts a compatibility measure of $o_t$ and $h_t$ using a learned neural network $f_{\mathrm{obs}}(o_t, h_t)$.

the Bayes filter tracks the belief explicitly, DVRL performs much better than generic RNNs under partial observability. However, a Bayes filter normally assumes a generative observation model, that defines the probability $p(o \mid h_t)$ of receiving an observation $o = o_t$ given the latent state $h_t$ (Fig. 1b). Learning this model can be very challenging since it requires modeling all observation features, including features irrelevant for RL. When $o$ is an image, $p(o \mid h_t)$ is a distribution over all possible images. This means, e.g., to navigate in a previously unseen environment, we need to learn the distribution of all possible environments with their visual appearance, lighting condition, etc. — a much harder task than learning to extract features relevant to navigation, e.g., the traversable space.

We introduce the *Discriminative Particle Filter Reinforcement Learning* (DPFRL), a POMDP RL method that learns to explicitly track a belief over the latent state without a generative observation model, and make decisions based on features of the belief (Fig. 1a). DPFRL approximates the belief by a set of weighted learnable latent particles $\{(h_t^i, w_t^i)\}_{i=1}^K$, and it tracks this particle belief by a non-parametric Bayes filter algorithm, an *importance weighted particle filter*, encoded as a differentiable computational graph in the neural network architecture. The importance weighted particle filter applies *discriminative update* to the belief with an observation-conditioned transition model and a discriminative state-observation compatibility function (serving as the importance weights), both of which are learnable neural networks trained end-to-end. By using these update functions instead of the transition and observation models of the standard particle filter, DPFRL sidesteps the difficulty of learning a generative observation model (Fig. 1b). The model is discriminative in the sense that the compatibility function, $f_{\mathrm{obs}}(o_t, h_t)$, as shown in Fig. 1c, while playing an analogue role as $p(o_t \mid h_t)$, is not required to directly represent a normalized distribution over observations; and through end-to-end training it only needs to model observation features relevant for the RL task. Finally, to summarize the particle belief for the policy, we introduce novel learnable features based on *Moment-Generating Functions* (MGFs) (Bulmer, 1979). MGF features are computationally efficient and permutation invariant, and they can be directly optimized to provide useful higher-order moment information for learning a policy. MGF features could be also used as learned features of any empirical distribution in applications beyond RL.

We evaluate DPFRL on a range of POMDP RL domains: a continuous control task from Igl et al. (2018), Flickering Atari Games (Hausknecht & Stone, 2015), Natural Flickering Atari Games, a new domain with more complex observations that we introduce, and the Habitat visual navigation domain using real-world data (Savva et al., 2019). DPFRL outperforms state-of-the-art POMDP RL methods in most cases. Results show that belief tracking with a particle filter is effective for handling partial observability, and the discriminative update and MGF-based belief features allow for complex observations.

## 2 Related Work

Real-world decision-making problems are often formulated as POMDPs. POMDPs are notoriously hard to solve; in the worst case, they are computationally intractable (Papadimitriou & Tsitsiklis, 1987). Approximate POMDP solvers have made dramatic progress in solving large-scale POMDPs (Kurniawati et al., 2008). Particle filters have been widely adopted as a belief tracker for POMDP solvers (Silver & Veness, 2010; Somani et al., 2013) having the flexibility to model complex and multi-modal distributions, unlike Gaussian and Kalman filters. However, predefined model and state representations are required for these methods (see e.g. Bai et al. (2015)).

Figure 2: **DPFRL Network.** In DPFRL, latent particles $\{(h_t^i, w_t^i)\}_{i=1}^K$ are maintained by a differentiable particle filter algorithm, which includes observation-conditioned transition function $f_{\text{trans}}$, discriminative compatibility function $f_{\text{obs}}$, and soft-resampling. The policy and value function is conditioned on the belief, which is summarized by the mean particle $\bar{h}_t$, and $m$ moment generating function features, $M_t^{1:m}$.

Given the advances in generative neural network models, various neural models have been proposed for belief tracking (Chung et al., 2015; Maddison et al., 2017; Le et al., 2018; Naesseth et al., 2018). DVRL (Igl et al., 2018) uses a Variational Sequential Monte-Carlo method (Naesseth et al., 2018), similar to the particle filter we use, for belief tracking in RL. This gives better belief tracking capabilities, but as we demonstrate in our experiments, generative modeling is not robust in complex observation spaces with high-dimensional irrelevant observation. More powerful generative models, e.g., DRAW (Gregor et al., 2015), could be considered to improve generative observation modeling; however, evaluating a complex generative model for each particle would significantly increase the computational cost and optimization difficulty.

Learning a robust latent representation and avoiding reconstructing observations are of great interest for RL (Oord et al., 2018; Guo et al., 2018; Hung et al., 2018; Gregor et al., 2019; Gelada et al., 2019). Discriminative RNNs have also been widely used for belief approximation in partially observable domains (Bakker, 2002; Wierstra et al., 2007; Foerster et al., 2016). The latent representation is directly optimized for the policy $p(a|h_t)$ that skips observation modeling. For example, Hausknecht & Stone (2015) and Zhu et al. (2018) tackle partially observable Flickering Atari Games by extending DQN (Mnih et al., 2013) with an LSTM memory. Our experiments demonstrate that the additional structure for belief tracking provided by a particle filter can give improved performance in RL.

Embedding algorithms into neural networks to allow end-to-end discriminative training has gained attention recently. For belief tracking, the idea has been used in the differentiable histogram filter (Jonschkowski & Brock, 2016), Kalman filter (Haarnoja et al., 2016) and particle filter (Karkus et al., 2018; Jonschkowski et al., 2018). Further, Karkus et al. (2017) combined a learnable histogram filter with the Value Iteration Network (Tamar et al., 2016) and introduced a learnable POMDP planner, QMDP-net. However, these methods require a predefined state representation and are limited to relatively small state spaces. Ma et al. (2019) integrated the particle filter with standard RNNs, e.g., the LSTM, and introduced PF-RNNs for sequence prediction. We build on the work of Ma et al. (2019) and demonstrate its advantages for RL with complex partial observations, and extend it with MGF features for improved decision making from particle beliefs. Note that our framework is not specific to PF-RNNs, and could be applied to other differentiable particle filters as well.

## 3 DISCRIMINATIVE PARTICLE FILTER REINFORCEMENT LEARNING

We introduce DPFRL for reinforcement learning under partial and complex observations. The DPFRL architecture is shown in Fig. 2. It has two main components, a discriminatively trained particle filter that tracks a latent belief $b_t$, and an actor network that learns a policy $p(a \mid b_t)$ given the belief $b_t$.

### 3.1 PARTICLE FILTER FOR LATENT BELIEF TRACKING

**Latent State Representation.** In POMDPs the semantics of states $s$ is typically defined explicitly. State variables may correspond to the position of a robot, configuration of obstacles, etc. In DPFRL, we do not require explicit specification of the state variables, but implicitly represent the state as a vector $h$ of latent variables, that is, the semantics of the state variables are learned instead of being pre-specified. We use a fully differentiable particle filter algorithm to maintain a belief over $h$. More specifically, we approximate the belief with a set of weighted latent particles $b_t \approx \{(h_t^i, w_t^i)\}_{i=1}^K$, where $\{h_t^i\}_{i=1}^K$ are $K$ latent states learned by policy-oriented training, and $\{w_t^i\}_{i=1}^K$ represents the

corresponding weights. Each latent state $h_t^i$ stands for a hypothesis in the belief; the set of latent particles provide an approximate representation for the belief.

**Belief Update.** In a basic particle filter, there are two key steps to update a particle belief $\{h_{t-1}^i, w_{t-1}^i\}_{i=1}^K$ to a new particle belief $\{h_t^i, w_t^i\}_{i=1}^K$ upon receiving an observation $o_t$ after executing action $a_t$.

$$h_t^i \sim p(h \mid h_{t-1}^i, a_t), \tag{1}$$

$$w_t^i = \eta p(o_t \mid h_t^i) w_{t-1}^i, \quad \eta = 1/\Sigma_{i=1}^K p(o_t \mid h_t^i) w_{t-1}^i \tag{2}$$

The first step, Eq. 1, takes the transition dynamics into account to update each particle. The second step, Eq. 2, takes the observation into account to reweigh the new particles.

Our belief update has a similar structure as the standard particle filter, but we replace the transition model and the observation model with richer functions to make the update more suitable for learning a policy in a partially observable domain. Specifically, the update equations are as follows.

$$h_t^i \sim f_{\text{trans}}(h_{t-1}^i, a_t, o_t), \tag{3}$$

$$w_t^i = \eta f_{\text{obs}}(h_t^i, o_t) w_{t-1}^i, \quad \eta = 1/\Sigma_{i=1}^K f_{\text{obs}}(h_t^i, o_t) w_{t-1}^i, \tag{4}$$

$$\{(h_t^{\prime i}, w_t^{\prime i})\}_{i=1}^K = \text{Soft-Resampling}(\{(h_t^i, w_t^i)\}_{i=1}^K) \tag{5}$$

Below, we first explain the intuition behind the above updates and the roles of $f_{\text{trans}}$ and $f_{\text{obs}}$ as compared to the standard transition and observation models. We then derive that above rules from an importance weighed particle filter in Sect. 3.2.

**Observation-conditioned transition update.** Eq. 3 takes a form more general than that in Eq. 2: instead of using the transition dynamics $p(h \mid h_{t-1}^i, a_t)$ to evolve a particle, we use a more general observation-conditioned transition $f_{\text{trans}}(h \mid h_{t-1}^i, a_t, o_t)$. Incorporating the observation allows alleviating the problem of sampling unlikely particles. In fact, if we take $f_{\text{trans}}$ to be $p(h \mid h_{t-1}^i, a_t, o_t)$, then this allows us to skip Eq. 2, and completely avoids sampling particles that are likely considering $a_t$ only, but unlikely considering both $a_t$ and $o_t$. Of course, in RL we do not have access to $p(h \mid h_{t-1}^i, a_t, o_t)$, and instead $f_{\text{trans}}$ is learned. In our implementation, a network first extracts features from $o_t$, they are fed to a gated function following the PF-GRU of Ma et al. (2019), which outputs the mean and variance of a normal distribution. Details are in the Appendix.

**Importance weighting via a compatibility function.** Eq. 4 is a relaxed version of Eq. 2: instead of using the observation model $p(o_t \mid h_t^i)$ to adjust the particle weights based on their compatibility with the observation, we use a general non-negative compatibility function $f_{\text{obs}}(h_t^i, o_t)$. If the compatibility function is required to satisfy the normalization constraint that $\sum_o f_{\text{obs}}(h, o)$ is a constant for all $h$, then it is equivalent to a conditional distribution of $o$ given $h$. We do not require this, and thus the update loses the probabilistic interpretation in Eq. 2. However, eliminating the need for the normalization constraint allows the compatibility function to be efficiently trained, as we can avoid computing the normalization constant. In addition, since the observation has already been incorporated in Eq. 3, we actually expect that the weights need to be adjusted in a way different from the standard particle filter. In our implementation, $f_{\text{obs}}(h_t^i, o_t)$ is a neural network with a single fully connected layer that takes in a concatenation of $h_t^i$ and features extracted from $o_t$. The output of the network is interpreted as the log of $f_{\text{obs}}$; and for numerical stability we perform the weight updates of Eq. 4 in the log-space as well. Note that more complex network architectures could improve the capability of $f_{\text{obs}}$, which we leave to future work.

**Soft-resampling.** To avoid particle degeneracy, i.e., most of the particles having a near-zero weight, particle filters typically resample particles. We adopt the soft-resampling strategy of Karkus et al. (2018); Ma et al. (2019), that provides approximate gradients for the non-differentiable resampling step. Instead of sampling from $p_t(i) = w_t^i$, we sample particles $\{h_t^{\prime i}\}_{i=1}^K$ from a softened proposal distribution $q(i) = \alpha w_t^i + (1 - \alpha)1/K$, where $\alpha$ is an trade-off parameter. The new weights are derived using importance sampling: $w_t^{\prime i} = \frac{w_t^i}{\alpha w_t^i + (1-\alpha)1/K}$. We can have the final particle belief as $\{(h_t^{\prime i}, w_t^{\prime i})\}_{i=1}^K = \text{Soft-Resampling}(\{(h_t^i, w_t^i)\}_{i=1}^K)$. As a result, $f_{\text{obs}}$ can be optimized with global belief information and model shared useful features across multiple time steps.

Another related concern is that the particle distribution may collapse to particles with the same latent state. This can be avoided by ensuring that the stochastic transition function $f_{\text{trans}}$ has a non-zero variance, e.g., by adding a small constant to the learned variance.

**End-to-end training.** In DPFRL the observation-conditioned transition function $f_{\text{trans}}$ and the compatibility function $f_{\text{obs}}$ are learned. Instead of training for a modeling objective, they are trained end-to-end for the final RL objective, backpropagating gradients through the belief-conditional policy $p(a \mid b_t)$ and the update steps of the particle filter algorithm, Eq. 3-5.

## 3.2 Connection to Importance Weighted Particle Filter

Our belief update can be motivated from the following importance weighted particle filter. Learning directly $p(h' \mid h, a, o)$ is generally difficult, but if we have a distribution $q(h' \mid h, a, o)$ that is easy to learn, then we can use importance sampling to update a particle belief.

$$h_t^i \sim q(h_{t-1}^i, a_t, o_t), \tag{6}$$

$$w_t^i = \eta f(h_t^i, h_{t-1}^i, a_t, o_t) w_{t-1}^i, \quad \eta = 1/\Sigma_{i=1}^K f(h_t^i, h_{t-1}^i, a_t, o_t) w_{t-1}^i \tag{7}$$

where $f = p/q$ is the importance weight.

Consider the case that $q(h' \mid h, a, o)$ is the conditional distribution of a joint distribution $q(h', h, a, o)$ of the form $p(h' \mid h, a)q(o \mid h')$. That is, $p$ and $q$ share the same transition dynamics $p(h' \mid h, a)$. Then the importance weight $f$ is a function of $h'$ and $o$ only, because

$$f(h', h, a, o) = \frac{p(h' \mid h, a, o)}{q(h' \mid h, a, o)} = \frac{p(h' \mid h, a)p(o \mid h')}{p(h' \mid h, a)q(o \mid h')} = \frac{p(o \mid h')}{q(o \mid h')}.$$

This simpler form is exactly the form that we used for $f_{\text{obs}}$ in our belief update.

## 3.3 Discriminative vs. Generative Modeling

We expect the discriminative compatibility function to be more effective than a generative model for the following reasons. A generative model aims to approximate $p(o \mid h)$ by learning a function that takes $h$ as input and outputs a parameterized distribution over $o$. When $o$ is, e.g., an image, this requires approximations, e.g., using pixel-wise Gaussians with learned mean and variance. This model is also agnostic to the RL task and considers all observation features equally, including features irrelevant for filtering and decision making.

In contrast, $f_{\text{obs}}$ takes $o$ and $h$ as inputs, and estimates the compatibility of $o$ and $h$ for particle filtering directly. This function avoids forming a parametric distribution over $o$, and the function can be easier to learn. The same functional form is used for the energy-function of energy-based models (LeCun et al., 2006) and in contrastive predictive coding (Oord et al., 2018), with similar benefits. For example, $f_{\text{obs}}$ may learn unnormalized likelihoods that are only proportionate to $p(o \mid h)$ up to a $o$-dependent value, because after the normalization in Eq. 4, they would give the same belief update as the normalized $p(o \mid h)$. Further, because $f_{\text{obs}}$ is trained for the final RL objective instead of a modeling objective, it may learn a compatibility function that is useful for decision making, but that does not model all observation features and has no proper probabilistic interpretation.

While the task-oriented training of discriminative models may improve policy performance for the reasons above, it cannot take advantage of an auxiliary learning signal like the reconstruction objective of a generative model. An interesting line of future work may combine generative models with a compatibility function to simultaneously benefit from both formulations.

## 3.4 Belief-Conditional Actor Network

Conditioning a policy directly onto a particle belief is non-trivial. To feed it to the networks, we need to summarize it into a single vector.

We introduce a novel feature extraction method for empirical distributions based on *Moment-Generating Functions* (MGFs). The MGF of an $n$-dimensional random variable $\mathbf{X}$ is given by $M_{\mathbf{X}}(\mathbf{v}) = \mathbb{E}[e^{\mathbf{v}^\top \mathbf{X}}], \mathbf{v} \in \mathbb{R}^n$. In statistics, MGF is an alternative specification of its probability distribution (Bulmer, 1979). Since particle belief $b_t$ is an empirical distribution, the moment generating function of $b_t$ can be denoted as $M_{b_t}(\mathbf{v}) = \sum_{i=1}^K w_t^i e^{\mathbf{v}^\top h_t^i}$. A more detailed background on MGFs is in Appendix A.2.

In DPFRL, we use the values of the MGF at $m$ *learned* locations $\mathbf{v}^{1:m}$ as the feature vector of the MGF. The $j$-th MGF feature is given by $M_{b_t}^j(\mathbf{v}^j)$. For a clean notation, we use $M_t^j$ in place of $M_{b_t}^j(\mathbf{v}^j)$. We use $\left[\bar{h}_t, M_t^{1:m}\right]$ as features for belief $b_t$, where $\bar{h}_t = \sum_{i=1}^K w_t^i h_t^i$ is the mean particle. The mean particle $\bar{h}_t$, as the first-order moment, and $m$ additional MGF features, give a summary of the belief characteristics. The number of MGF features, $m$, controls how much additional information we extract from the belief. We empirically study the influence of MGF features in ablation studies.

Compared to Ma et al. (2019) that uses the mean as the belief estimate, MGF features provide additional features from the empirical distribution. Compared to DVRL (Igl et al., 2018) that treats the Monte-Carlo samples as a sequence and merges them by an RNN, MGF features are permutation-invariant, computationally efficient and easy to optimize, especially when the particle set is large.

Given the features $\left[\bar{h}_t, M_t^{1:m}\right]$ for $b_t$, we compute the policy $p(a \mid b_t)$ with a policy network $\pi(b_t)$. We trained with an actor-critic RL algorithm, A2C (Mnih et al., 2016), where a value network $V(b_t)$ is introduced to assist learning. We use small fully-connected networks for $\pi(b_t)$ and $V(b_t)$ that share the same input $b_t$.

## 4 EXPERIMENTS

We evaluate DPFRL in a range of POMDP RL domains with increasing belief tracking and observation modeling complexity. We first use benchmark domains from the literature, Mountain Hike, and 10 different Flickering Atari Games. We then introduce a new, more challenging domain, Natural Flickering Atari Games, that uses a random video stream as the background. Finally we apply DPFRL to a challenging visual navigation domain with RGB-D observations rendered from real-world data.

We compare DPFRL with a GRU network, a state-of-the-art POMDP RL method, DVRL, and ablations of the DPFRL architecture. As a brief conclusion, we show that: 1) DPFRL significantly outperforms GRU in most cases because of its explicit structure for belief tracking; 2) DPFRL outperforms the state-of-the-art DVRL in most cases even with simple observations, and its benefit increases dramatically with more complex observations because of DPFRL's discriminative update; 3) MGF features are more effective for summarizing the latent particle belief than alternatives.

### 4.1 EXPERIMENTAL SETUP

We train DPFRL and baselines with the same A2C algorithm, and use a similar network architecture and hyperparameters as the original DVRL implementation. DPFRL and DVRL differ in the particle belief update structure, but they use the same latent particle size $\dim(h)$ and the same number of particles $K$ as in the DVRL paper ($\dim(h) = 128$ and $K = 30$ for Mountain Hike, $\dim(h) = 256$ and $K = 15$ for Atari games and visual navigation). The effect of the number of particles is discussed in Sect. 4.5. We train all models for the same number of iterations using the RMSProp optimizer (Tieleman & Hinton, 2012). Learning rates and gradient clipping values are chosen based on a search in the BeamRider Atari game independently for each model. Further details are in the Appendix. We have not performed additional searches for the network architecture and other hyper-parameters, nor tried other RL algorithm, such as PPO (Schulman et al., 2017), which may all improve our results.

All reported results are averages over 3 different random seeds. We plot rewards accumulated in an episode, same as DVRL (Igl et al., 2018). The curves are smoothed over time and averaged over parallel environment executions.

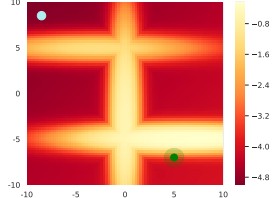

Figure 3: **Mountain Hike Task**. An agent navigates on the map from the start position (white dot) to the goal (green dot with the shaded area as the threshold). Partial observation is introduced by a Gaussian noise and appended with a long noise vector of length $l$. The reward $r(x, y)$ for position $(x, y)$ is given by the heat map.

### 4.2 MOUNTAIN HIKE

Mountain Hike was introduced by Igl et al. (2018) to demonstrate the benefit of belief tracking for POMDP RL. It is a continuous control problem where an agent navigates on a fixed $20 \times 20$ map.

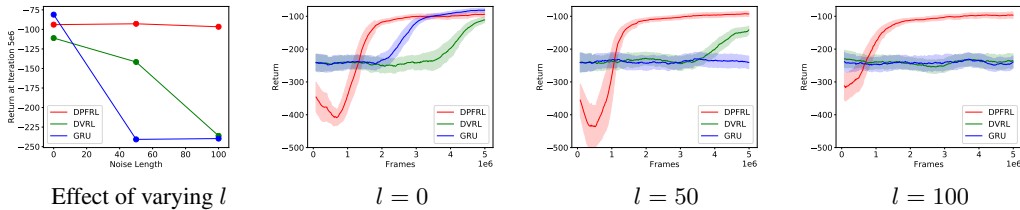

| Effect of varying $l$ | $l = 0$ | $l = 50$ | $l = 100$ |

Figure 4: Results for Mountain Hike where useful observations are concatenated with a noise vector of length $l$.

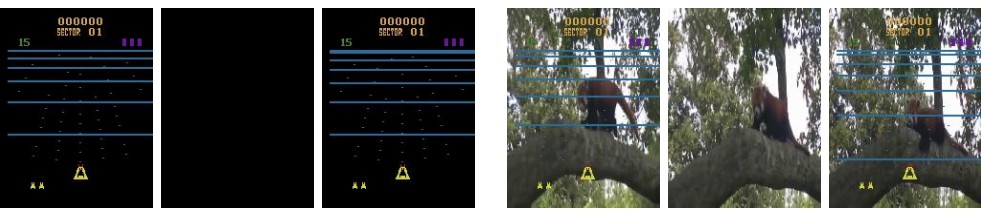

Flickering Atari Games — Natural Flickering Atari Games

Figure 5: **Partially Observable Atari Games.** In Flickering Atari Games frames are randomly dropped and replaced with a blank frame. In Natural Flickering Atari Games the background is replaced with a random video stream and the Atari components of the image are randomly dropped.

In the original task, partial observability is introduced by disturbing the agent observation with an additive Gaussian noise. To illustrate the effect of observation complexity in natural environments, we concatenate the original observation vector with a random noise vector. The complexity of the optimal policy remains unchanged, but the relevant information is now coupled with irrelevant observation features. More specifically, the state space and action space in Mountain Hike are defined as $\mathcal{S} = \mathcal{A} = \mathbb{R}^2$, where $s_t = [x_t, y_t]$ and $a_t = [\delta x_t, \delta y_t]$. Transitions of the agent are stochastic with an additive Gaussian noise: $s_{t+1} = s_t + a_t + \epsilon_a$, where $\epsilon_a \sim \mathcal{N}(0, 0.25)$. The observation space is $\mathcal{O} = \mathbb{R}^{2+l}$, where $l$ is a predefined constant and $l = 0$ corresponds to the original setting. Observations are $o_t = [o_t^s, o_t^n]$, where $o_t^s = s_t + \epsilon_s, \epsilon_s \sim \mathcal{N}(0, 1)$, and $o_t^n \in \mathbb{R}^l$ is sampled from a uniform distribution $\mathcal{U}(-10, 10)$. The reward for each step is given by $r_t = r(x_t, y_t) - 0.01||a_t||$ where $r(x_t, y_t)$ is shown in Fig. 3. Episodes end after 75 steps. We train models for different settings of the noise vector length $l$, from $l = 0$ to $l = 100$. Results are shown in Fig. 4. We observe that DPFRL learns faster than the DVRL and GRU in all cases, including the original setting $l = 0$. Importantly, as the noise vector length increases, the performance of DVRL and GRU degrades, while DPFRL is unaffected. This demonstrates the ability of DPFRL to track a latent belief without having to explicitly model complex observations.

### 4.3 ATARI GAMES WITH PARTIAL OBSERVABILITY

Atari games are one of the most popular benchmark domains for RL methods (Mnih et al., 2013). Their partially observable variants, Flickering Atari Games, have been used to benchmark POMDP RL methods (Hausknecht & Stone, 2015; Zhu et al., 2018; Igl et al., 2018). Here image observations are single frames randomly replaced by a blank frame with a probability of 0.5. The flickering observations introduce a simple form of partial observability. Another variant, Natural Atari Games (Zhang et al., 2018), replaces the simple black background of the frames of an Atari game with a randomly sampled video stream. This modification brings the Atari domain one step closer to the visually rich real-world, in that the relevant information is now encoded in complex observations. As shown by Zhang et al. (2018), this poses a significant challenge for RL.

We propose a new RL domain, *Natural Flickering Atari Games*, that involves both challenges: partial observability simulated by flickering frames, and complex observations simulated by random background videos. The background videos increase observation complexity without affecting the decision making complexity, making this a suitable domain for evaluating RL methods with complex observations. We sample the background video from the ILSVRC dataset (Russakovsky et al., 2015). Examples for the BeamRider game are shown in Fig. 5. Details are in Appendix B.

Table 1: Results for Flickering Atari Games and Natural Flickering Atari Games

| | Flickering Atari Games | | | Natural Flickering Atari Games | | |
| | DPFRL | DVRL[Igl et al., 2018] | GRU[Igl et al., 2018] | DPFRL | DVRL | GRU |
|---|---|---|---|---|---|---|
| Pong | 15.40±0.76 | **18.17±2.67** | 6.33±3.03 | **15.65±1.99** | -19.78±0.06 | 2.62±0.93 |
| ChopperCommand | **8,086±159.1** | 6,602±449 | 5,150± 488.1 | **1,566±67.03** | 1,068±128.9 | 1,418±5.08 |
| MsPacman | **3,028±545.3** | 2,221±199 | 2,312±358 | **2,106±123.9** | 1,358±155.4 | 1,833±45.45 |
| Centipede | **4,849±291.4** | 4,240±116 | 4,395±224 | **4,164±23.0** | 3,154±335.9 | 3,679±116.4 |
| BeamRider | **3,940±107.4** | 1,663±183 | 1,801±65 | **682.9±37.42** | 437.3±46.31 | 525.6±25.25 |
| Frostbite | 293.5±5.06 | 297.0±7.85 | 254.0±0.45 | **260.2±4.60** | 252.4±6.48 | 254.3±9.20 |
| Bowling | **33.89±0.34** | 29.53 ±0.23 | 30.0±0.18 | **29.45±0.13** | 24.80±0.31 | 27.13±0.41 |
| IceHockey | **-4.06±0.02** | -4.88±0.17 | -7.10±0.60 | -6.08±0.18 | -8.79±0.12 | -5.30±0.66 |
| DDunk | -11.25±1.25 | **-5.95±1.25** | -15.88±0.34 | -15.36±0.96 | -17.62±0.16 | -14.31±0.37 |
| Asteroids | **1,948±202.6** | 1,539±73 | 1,545±51 | 1,489±15.76 | 1,406±132.3 | 1,675±571.5 |

We evaluate DPFRL for both Flickering Atari Games and Natural Flickering Atari Games. We use the same set of games as Igl et al. (2018). To ensure a fair comparison, we take the GRU and DVRL results from the paper for Flickering Atari Games, use the same training iterations as in Igl et al. (2018), and we use the official DVRL open source code to train for Natural Flickering Atari Games. Results are summarized in Table 1. We highlight the best performance in bold where the difference is statistically significant ($p = 0.05$). Detailed training curves are in Appendix E.

We observe that DPFRL significantly outperforms GRU in almost all games, which indicates the importance of explicit belief tracking, and shows that DPFRL can learn a useful latent belief representation. Despite the simpler observations, DPFRL significantly outperforms DVRL and achieves state-of-the-art results on 5 out of 10 standard Flickering Atari Games (ChopperCommand, MsPacman, BeamRider, Bowling, Asteroids), and it performs comparably in 3 other games (Centipede, Frostbite, IceHockey). The strength of DFPRL shows even more clearly in the Natural Flickering Atari Games, where it significantly outperforms DVRL on 7 out of 10 games and performs similarly in the rest. In some games, e.g. in Pong, DPFRL performs similarly with and without videos in the background (15.65 vs. 15.40), while the DVRL performance degrades substantially (-19.78 vs. 18.17). These results show that while the architecture of DPFRL and DVRL are similar, the policy-oriented discriminative update of DPFRL is much more effective for handling complex observations, and the MGF features provide a more powerful summary of the particle belief for decision making. However, on some games, e.g. on ChopperCommand, even DPFRL performance drops significantly when adding background videos. This shows that irrelevant features can make a task much harder, even for a discriminative approach, as also observed by Zhang et al. (2018).

## 4.4 VISUAL NAVIGATION

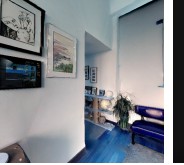 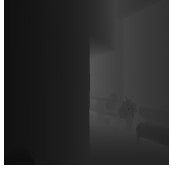

Figure 6: RGB-D Habitat Observations

Table 2: Visual Navigation Results

| | SPL | Success Rate | Reward |
|---|---|---|---|
| DPFRL | **0.79** | **0.88** | **12.82±5.82** |
| DVRL | 0.09 | 0.11 | 5.22±2.24 |
| GRU | 0.63 | 0.74 | 10.14±2.82 |
| PPO[Savva et al., 2019] | 0.70 | 0.80 | — |

Visual navigation poses a great challenge for deep RL (Mirowski et al., 2016; Zhu et al., 2017; Lample & Chaplot, 2017). We evaluate DPFRL for visual navigation in the Habitat Environment (Savva et al., 2019), using the real-world Gibson dataset (Xia et al., 2018). In this domain, a robot needs to navigate to goals in previously unseen environments. In each time step, it receives a first-person RGB-D camera image and its distance and relative orientation to the goal. The main challenge lies in the partial and complex observations: first-person view images only provide partial information about the unknown environment; and the relevant information for navigation, traversability, is encoded in rich RGB-D observations along with many irrelevant features, e.g., the texture of the wall. We use the Gibson dataset with the training and validation split provided by the Habitat challenge.

We train models with the same architecture as for the Atari games, except for the observation encoder that accounts for the different observation format. We evaluate models in unseen environments from

Table 3: Ablation Study Results for the Natural Flickering Atari Games

| Envs | DPFRL | DPFRL-generative | DPFRL-P1 | DPFRL-mean | DPFRL-GRUmerge |
|---|---|---|---|---|---|
| Pong | **15.65±1.99** | -20.21±0.02 | -18.60±0.08 | -5.53±14.35 | 13.14±4.01 |
| ChopperCommand | **1,566±67.03** | 1,027± 12.94 | 1,287±255.0 | 1,091±109.9 | 1,530±29.31 |
| MsPacman | 2,106±123.9 | 2,130±182.3 | **2,233±0.47** | 1,878±63.86 | 1,930±48.54 |
| Centipede | **4,164±23.0** | 3,194±339.4 | 3,557±398.1 | 3,599±439.8 | 4,093±76.4 |
| BeamRider | **682.9±37.42** | 498.1±8.38 | 570.7±172.3 | 645.5±227.4 | 603.8±40.25 |
| Frostbite | **260.2±4.60** | 255.9±2.78 | 178.2±81.5 | 178.4±81.70 | 252.1±0.48 |
| Bowling | 29.45±0.13 | 24.68±0.13 | 25.94±0.55 | 26.0±0.81 | **29.50±0.33** |
| IceHockey | -6.08±0.18 | -7.88±0.30 | -6.02±1.03 | -6.25±1.96 | -5.85±0.30 |
| DDunk | -15.36±0.96 | -15.59±0.06 | -13.28±0.96 | -14.42±0.18 | -14.39±0.24 |
| Asteroids | 1,406±132.3 | 1,415±5.33 | **1,618±64.45** | 1,433±40.73 | 1,397±11.44 |

the validation split and compute the same metrics as in the literature: SPL, success rate, and average rewards. Results are shown in Table 2. Further details and results are in Appendix B and E.

DPFRL significantly outperforms both DVRL and GRU in this challenging domain. DVRL performs especially poorly, demonstrating the difficulty of learning a generative observation model in realistic, visually rich domains. DPFRL also outperforms the PPO baseline from Savva et al. (2019).

We note that submissions to the recently organized Habitat Challenge 2019 (Savva et al., 2019), such as (Chaplot et al., 2019), have demonstrated better performance than the PPO baseline (while our results are not directly comparable because of the closed test set of the competition). However, these approaches rely on highly specialized structures, such as 2D mapping and 2D path planning, while we use the same generic network as for Atari games. Future work may further improve our results by adding a task-specific structure to DPFRL or training with PPO instead of A2C.

## 4.5 ABLATION STUDY

We conduct an extensive ablation study on the Natural Flickering Atari Games to understand the influence of each DPFRL component. The results are presented in Table 3.

*The discriminative compatibility function is more effective than a generative observation function.* DPFRL-generative replaces the discriminative compatibility function of DPFRL with a generative observation function, where grayscale image observations are modeled by pixel-wise Gaussian distributions with learned mean and variance. Unlike DVRL, DPFRL-generative only differs from DPFRL in the parameterization of the observation function, the rest of the architecture and training loss remains the same. In most cases, the performance for DPFRL-generative degrades significantly compared to DPFRL. These results are aligned with our earlier observations, and indicate that the compatibility function is capable of extracting the relevant information from complex observations without having to learn a more complex generative model.

*More particles perform better.* DPFRL with 1 particle performs poorly on most of the tasks (DPFRLL-P1). This indicates that a single latent state is insufficient to represent a complex latent distribution that is required for the task, and that more particles may improve performance.

*MGF features are useful.* We compare DPFRL using MGF features with DPFRL-mean, that only uses the mean particle; and with DPFRL-GRUmerge, that uses a separate RNN to summarize the belief, similar to DVRL. Results show that DPFRL-mean does not work as well as the standard DPFRL, especially for tasks that may need complex belief tracking, e.g., Pong. This can be attributed to the more rich belief statistics provided by MGF features, and that they do not constrain the learned belief representation to be always meaningful when averaged. Comparing to DPFRL-GRUmerge shows that MGF features generally perform better. While an RNN may learn to extract useful features from the latent belief, optimizing the RNN parameters is harder, because they are not permutation invariant to the set of particles and they result in a long backpropagation chain.

## 5 CONCLUSION

We have introduced DPFRL, a framework for POMDP RL in natural environments. DPFRL combines the strength of Bayesian filtering and end-to-end RL: it performs explicit belief tracking with

learnable particle filters optimized directly for the RL policy. DPFRL achieved state-of-the-art results on POMDP RL benchmarks from prior work, Mountain Hike and a number of Flickering Atari Games. Further, it significantly outperformed alternative methods in a new, more challenging domain, Natural Flickering Atari Games, as well as for visual navigation using real-world data. We have proposed a novel MGF feature for extracting statistics from an empirical distribution. MGF feature extraction could be applied beyond RL, e.g., for general sequence prediction.

DPFRL does not perform well in some particular cases, e.g., DoubleDunk. While our task-oriented discriminative update are less susceptible to complex and noisy observations than a generative model, they do not benefit from an additional learning signal that could improve sample efficiency, e.g., through a reconstruction loss. Future work may combine a generative observation model with the discriminative update in the DPFRL framework.

## 6 ACKNOWLEDGEMENT

This research is partially supported by ONR Global and AFRL grant N62909-18-1-2023. We want to thank Maximilian Igl for suggesting to add videos to the background of Atari games.

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

# A    BACKGROUND

## A.1    PARTICLE FILTER ALGORITHM

Particle filter is an approximate Bayes filter algorithm for belief tracking. Bayes filters estimate the belief $b_t$, i.e., a posterior distribution of the state $s_t$, given the history of actions $a_{1:t}$ and observations $o_{1:t}$. Instead of explicitly modeling the posterior distribution, particle filter approximates the posterior with a set of weighted particles, $b_t \approx \{(s_t^i, w_t^i)\}_{i=1}^K$, and update the particles in a Bayesian manner. Importantly, the particle set could approximate arbitrary distributions, e.g., Gaussians, continuous multi-modal distributions, etc. The mean state can be estimated as the mean particle $\bar{s}_t = \sum_{i=1}^K w_t^i s_t^i$. The particle updates include three steps: transition update, measurement update, and resampling.

**Transition update.** We first update the particles by a given motion model. More specifically, we sample the next state $s_{t+1}^i$ from a generative transition function

$$s_{t+1}^i \sim p(s \mid s_t^i, a_t) \tag{8}$$

where $p(s \mid s_t^i, a_t)$ is the transition function.

**Measurement update.** The particle weights are updated again using the observation likelihoods

$$w_{t+1}^i = \eta p(o_t \mid s_{t+1}^i) w_t^i, \quad \eta = 1 / \sum_{i=1}^K w_{t+1}^i \tag{9}$$

where $\eta$ is a normalization factor and $p(o_t \mid s_{t+1}^i)$ is the observation likelihood computed by evaluating observation $o_t$ in a generative observation function $p(o \mid s_{t+1}^i)$.

**Resampling.** The particle filter algorithm can suffer from particle degeneracy, where after some update steps only a few particles have non-zero weights. This would prevent particle filter to approximate the posterior distribution effectively. Particle degeneracy is typically addressed by performing resampling, where new particles are sampled with repetition proportional to its weight. Specifically, we sample particles from a categorical distribution $p$ parameterized by the particle weights $\{w_t^i\}_{i=1}^K$

$$p(i) = w_t^i \tag{10}$$

where $p(i)$ is the probability for the $i$-th category, i.e., the $i$-th particle. The new particles approximate the same distribution, but they assign more representation capacity to the relevant regions of the state space.

## A.2    MOMENT-GENERATING FUNCTIONS

In probability theory, the moment-generating function (MGF) is an alternative specification of the probability distribution of a real-valued random variable (Bulmer, 1979). As its name suggests, MGF of a random variable could be used to generate any order of its moments, which characterize its probability distribution.

Mathematically, the MGF of a random variable $\mathbf{X}$ with dimension $m$ is defined by

$$M_{\mathbf{X}}(\mathbf{v}) = \mathbb{E}\left[e^{\mathbf{v}^\top \mathbf{X}}\right] \tag{11}$$

where $\mathbf{v} \in \mathbb{R}^m$ and we could consider the MGF of random variable $\mathbb{X}$ is the expectation of the random variable $e^{\mathbf{v}^\top \mathbf{X}}$.

Consider the series expansion of $e^{\mathbf{v}^\top \mathbf{X}}$

$$e^{\mathbf{v}^\top \mathbf{X}} = 1 + \mathbf{v}^\top \mathbf{X} + \frac{(\mathbf{v}^\top \mathbf{X})^2}{2!} + \ldots + \frac{(\mathbf{v}^\top \mathbf{X})^n}{n!} + \ldots \tag{12}$$

This leads to the well-known fact that the $j$-th order moment $M_j$ ($j$-way tensor) is the $j$-th order derivative of the MGF at $\mathbf{v} = 0$.

$$M_j = \frac{d^j M_{\mathbf{X}}}{d\mathbf{v}^j} \Big|_{\mathbf{v}=0} \tag{13}$$

In DPFRL, we use MGFs as additional features to provide moment information of the particle distribution. DPFRL learns to extract useful moment features for decision making by directly optimizing for policy $p(a \mid b_t)$.

## B  EXPERIMENT DETAILS

### B.1  IMPLEMENTATION DETAILS

**Observation Encoders:** For the observation encoders, we used the same structure with DVRL (Igl et al., 2018) for a fair comparison. For Mountain Hike, we use two fully connected layers with batch normalization and ReLU activation as the encoder. The dimension for both layers is 64. For the rest of the domains, we first down-sample the image size to $84 \times 84$, then we process images with 3 2D-convolution layers with channel number (32, 64, 32), kernel sizes (8, 4, 3) and stride (4, 2, 1), without padding. The compass and goal information are a vector of length 2; they are appended after the image encoding as the input.

**Observation Decoders:** Both DVRL and PFGRU-generative need observation decoders. For the Mountain Hike, we use the same structure as the encoder with a reversed order. The transposed 2D-convolutional network of the decoder has a reversed structure. The decoder is processed by an additional fully connected layer which outputs the required dimension (1568 for Atari and Habitat Navigation, both of which have $84 \times 84$ observations).

**Observation-conditioned transition network:** We directly use the transition function in PF-GRU (Ma et al., 2019) for $f_{\text{trans}}(h_{t-1}^i, a_t, o_t)$, which is a stochastic function with GRU gated structure. Action $a_t$ is first encoded by a fully connected layer with batch normalization and ReLU activation. The encoding dimension for Mountain Hike is 64 and 128 for all the rest tasks. The mean and variance of the normal distribution are learned again by two additional fully connected layers; for the variance, we use Softplus as the activation function.

**State-observation compatibility network:** $f_{\text{obs}}$ is implemented by a single fully connected layer without activation. In DVRL, the observation function is parameterized over the full observation space $o$ and $p(o \mid h_{t-1}^i, a_t^i)$ is assumed as a multivariate independent Bernoulli distribution whose parameters are again determined by a neural network (Igl et al., 2018). For numerical stability, all the probabilities are stored and computed in the log space and the particle weights are always normalized after each weight update.

**Soft-resampling:** The soft-resampling hyperparameter $\alpha$ is set to be 0.9 for Mountain Hike and 0.5 for the rest of the domains. Note that the soft-resampling is used only for DPFRL, not including DVRL. DVRL averages the particle weights to $1/K$ after each resampling step, which makes the resampling step cannot be trained by the RL.

**Belief Summary:** The GRU used in DVRL and DPFRL-GRUmerge is a single layer GRU with input dimension equals the dimension of the latent vector plus 1, which is the corresponding particle weight. The dimension of this GRU is exactly the dimension of the latent vector. For the MGF features, we use fully connected layers with feature dimensions as the number of MGF features. The activation function used is the exponential function. We could potentially explore the other activation functions to test the generalized-MGF features, e.g., ReLU.

**Actor Network and Policy Network:** The actor network and policy network are two fully connected layers, which take in the belief summary $b_t = \left[ \bar{h}_t, M_t^{1:m} \right]$ as input. The output dimension of these two networks are chosen according to the RL tasks.

**Model Learning:** For RL, we use an A2C algorithm with 16 parallel environments for both Mountain Hike and Atari games; for Habitat Navigation, we only use 6 parallel environments due to the GPU memory constraints. The loss function for DPFRL and GRU-based policy is just the standard A2C loss, $L_t^{\text{A2C}} = \mathcal{L}_t^A + \lambda^V \mathcal{L}_t^V + \lambda^H \mathcal{L}_t^H$, where $\mathcal{L}_t^A$ is the policy loss, $\mathcal{L}_t^V$ is the value loss, $\mathcal{L}_t^H$ is the entropy loss for encouraging exploration, and $\lambda^V$ and $\lambda^H$ are two hyperparameters. For all experiments, we use $\lambda_V = 0.5$ and $\lambda^H = 0.01$. For DVRL, an additional encoding loss $L_t^E$ is used to train the sequential VAE, which gives a loss function $L_t^{\text{DVRL}} = L_t^{\text{A2C}} + \lambda^E \mathcal{L}_t^E$. We follow the default setting provided by Igl et al. (2018) and set $\lambda^E = 0.1$. The rest of the hyperparameters, including learning rate, gradient clipping value and $\alpha$ in soft-resampling are tuned according to the BeamRider and directly applied to all domains due to the highly expensive experiment setups. The learning rate for all the networks are searched among the following values: $(3 \times 10^{-5}, 5 \times 10^{-5}, 1 \times 10^{-4}, 2 \times 10^{-4}, 3 \times 10^{-4})$; the gradient clipping value are searched among $\{0.5, 1.0\}$; the soft-resampling $\alpha$ is searched among $\{0.5, 0.9\}$. The best performing learning rates were $1^{-4}$ for DPFRL and GRU, and $2^{-4}$ for DVRL;

the gradient clipping value for all models was 0.5; the soft-resampling $\alpha$ is set to be 0.9 for Mountain Hike and 0.5 for Atari games.

## B.2 Experimental Setup

**Natural Flickering Atari games** We follow the setting of the prior works (Zhu et al., 2018; Igl et al., 2018): 1) 50% of the frames are randomly dropped 2) a frameskip of 4 is used 3) there is a 0.25 chance of repeating an action twice. In our experiments, we sample background videos from the ILSVRC dataset (Russakovsky et al., 2015). Only the videos with the length longer than 500 frames are sampled to make sure the video length is long enough to introduce variability. For each new episode, we first sample a new video from the dataset, and a random starting pointer is sampled in this video. Once the video finishes, the pointer is reset to the first frame (not the starting pointer we sampled) and continues from there. **Experiment platform:** We implement all the models using PyTorch (Paszke et al., 2017) with CUDA 9.2 and CuDNN 7.1.2. Flickering Atari environments are modified based on OpenAI Gym (Brockman et al., 2016) and we directly use Habitat APIs for visual navigation. Collecting experience and performing gradient updates are done on a single computation node with access to one GPU. For Mountain Hike and Atari games we use NVidia GTX1080Ti GPUs. For Habitat visual navigation we use NVidia RTX2080Ti GPUs.

## C PF-GRU Network Architecture

We implement DPFRL with gated transition and observation functions for particle filtering similar to PF-GRU (Ma et al., 2019).

In standard GRU, the memory update is implemented by a gated function:

$$h_t = (1 - z_t) \circ \tanh(n_t) + z_t \circ h_{t-1}, \qquad n_t = W_n[r_t \circ h_{t-1}, x_t] + b_n \tag{14}$$

where $W_n$ and $b_n$ are the corresponding weights and biases, and $z_t \, r_t$ are the learned gates.

PF-GRU introduces stochastic cell update by assuming the update to the memory, $n_t^i$, follows a parameterized Gaussian distribution

$$n_t^i = W_n[r_t^i \circ h_{t-1}^i, x_t] + b_n + \epsilon_t^i, \quad \epsilon_t^i \sim \mathcal{N}(0, \Sigma_t^i), \quad \Sigma_t^i = W_\Sigma[h_{t-1}^i, x_t] + b_\Sigma \tag{15}$$

With $x_t = [f_{enc}^o(o_t), f_{enc}^a(a_t)]$, we implement the transition function $h_{t+1}^i \sim f_{trans}(h_t^i, o_t, a_t)$, where $f_{enc}^o$ is the encoding network for observation and $f_{enc}^a$ is the encoding network for the actions.

For the observation function, we directly use a fully connected layer $f_{obs}(h_t^i, o_t) = W_o[h_t^i, o_t] + b_o$, where $W_o$ and $b_o$ are the corresponding weights and biases.

## D DPFRL Algorithm

---
**Algorithm 1:** DPFRL

---
**Input:** Previous belief $b_{t-1} \approx \{(h_{t-1}^i, w_{t-1}^i)\}_{i=1}^K$, observation $o_t$, action $a_t$

$x_t^o \leftarrow \text{Encoder}(o_t)$             (encode the raw observation)

$h_t^i \sim f_{trans}(h_{t-1}^i, a_t, u_t(x_t^o))$             (transition update)

$w_t^i \leftarrow \eta f_{obs}(x_t^o, h_t^i) w_{t-1}^i, \quad \eta = 1/\sum\limits_{i=1}^K w_t^i$             (observation update)

$\{(h_t'^i, w_t'^i)\}_{i=1}^K \leftarrow \text{Soft-Resampling}(\{(h_t^i, w_t^i)\}_{i=1}^K)$             (soft-resampling)

$\bar{h}_t \leftarrow \sum\limits_{i=1}^K w_t'^i h_t'^i$             (compute the mean)

**for** $j = 1 : m$ **do**

     $M_t^j \leftarrow \sum\limits_{i=1}^K w_t'^i \exp(\mathbf{v}^j h_t'^i)$             (compute MGF features)

**end**

$p(a \mid b_t) \leftarrow \pi(\bar{h}_t, M_t^{1:m})$             (compute the policy)

$V(b_t) \leftarrow V(\bar{h}_t, M_t^{1:m})$             (compute the value)

**Output:** Updated belief $b_t \approx \{(h_t'^i, w_t'^i)\}_{i=1}^K$, policy $p(a \mid b_t)$ and value $V(b_t)$

---

# E ADDITIONAL RESULTS

## E.1 FLICKEIRNG ATARI GAMES PLOTS

We provide the accumulated reward curves for Atari experiments in this section.

**Standard Flickering Atari Games.** For the standard Flickering Atari Games we provide the training curves below. Results for DVRL and GRU are directly taken from Igl et al. (2018).

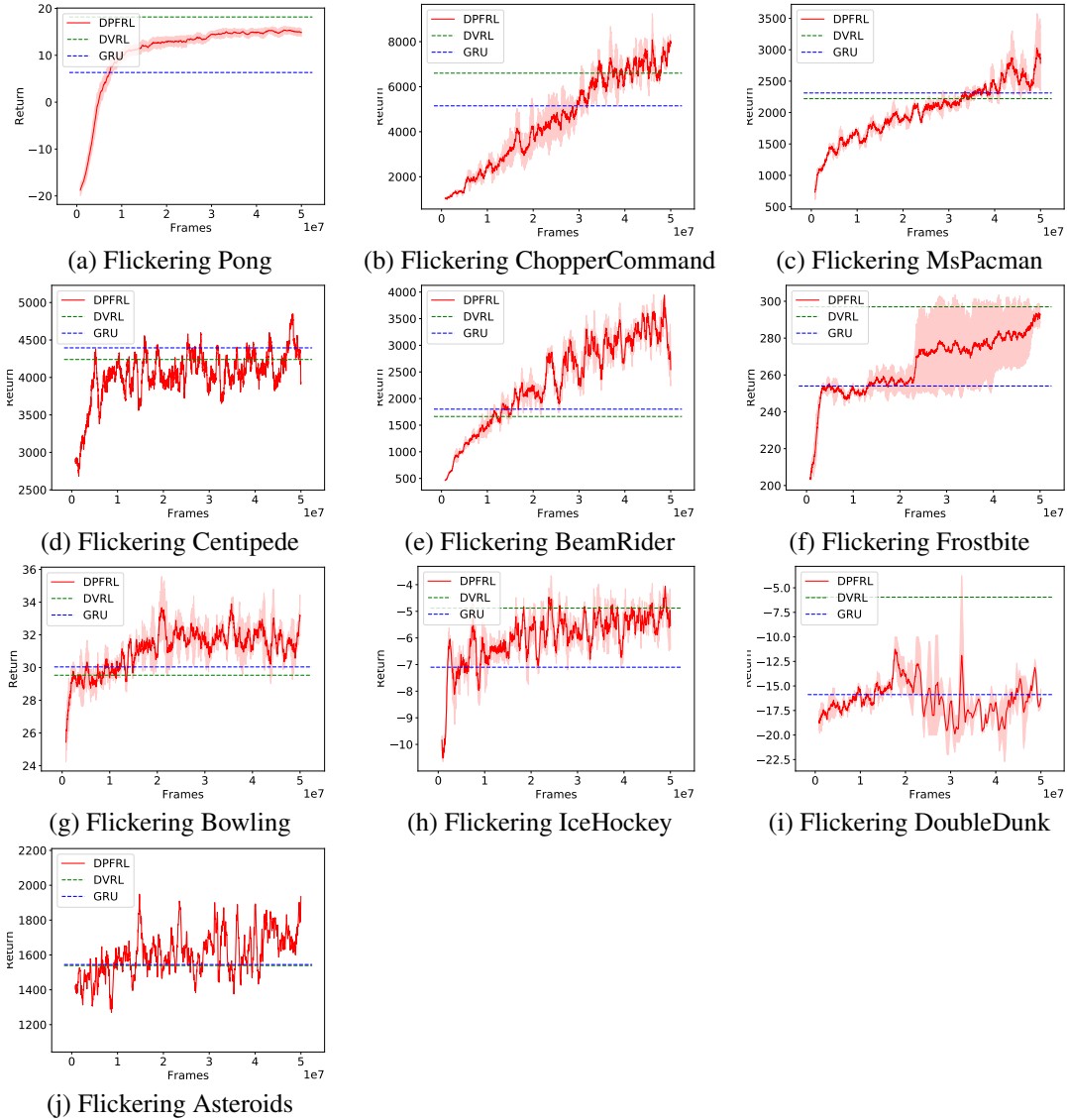

(a) Flickering Pong

(b) Flickering ChopperCommand

(c) Flickering MsPacman

(d) Flickering Centipede

(e) Flickering BeamRider

(f) Flickering Frostbite

(g) Flickering Bowling

(h) Flickering IceHockey

(i) Flickering DoubleDunk

(j) Flickering Asteroids

**Natural Flickering Atari Games.** For Natural Flickering Atari Games we report results for a separate validation set, where the background videos are different from the training set. The validation environment steps once after every 100 training iterations.

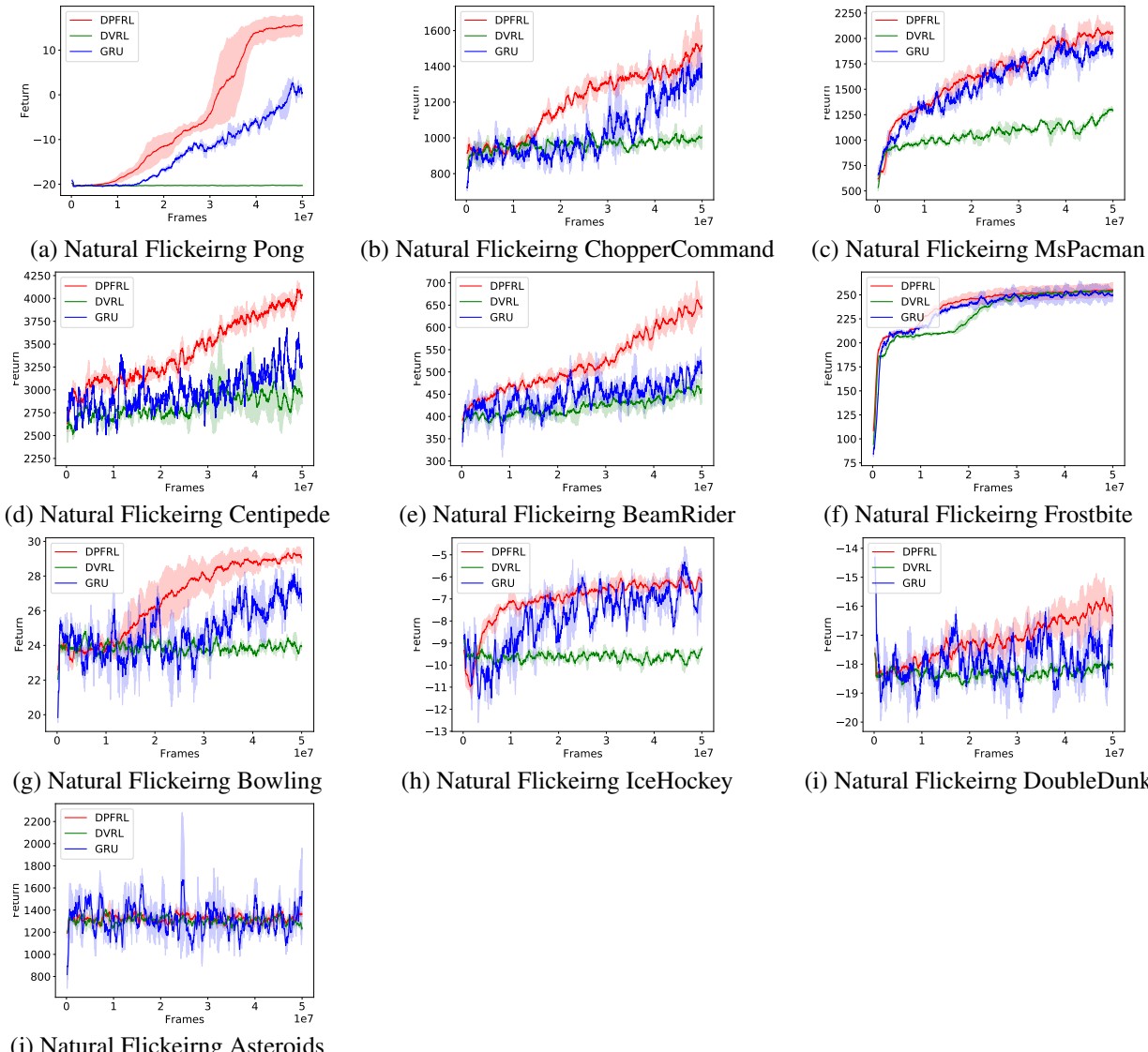

(a) Natural Flickeirng Pong   (b) Natural Flickeirng ChopperCommand   (c) Natural Flickeirng MsPacman

(d) Natural Flickeirng Centipede   (e) Natural Flickeirng BeamRider   (f) Natural Flickeirng Frostbite

(g) Natural Flickeirng Bowling   (h) Natural Flickeirng IceHockey   (i) Natural Flickeirng DoubleDunk

(j) Natural Flickeirng Asteroids

## E.2   VISUAL NAVIGATION

We present the reward curve for the Habitat visual navigation task below. DPFRL outperforms both GRU-based policy and DVRL given the same training time. DVRL struggles with training the observation model and fails during the first half of the training time. GRU based policy learns fast; given only the model-free belief tracker, it struggles to achieve higher reward after a certain point.

We only provide the reward curve here as SPL and success rate are only evaluated after the training is finished.

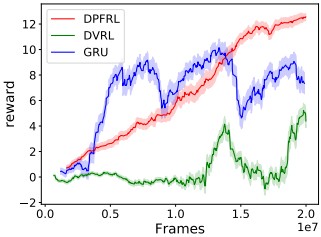

Figure 7: Habitat Visual Navigation Reward

## E.3   PARTICLE VISUALIZATION WITH PCA

We further visualize the latent particles by principal component analysis (PCA) and choose the first 2 components. We choose a trajectory in the Habitat Visual Navigation experiment, where 15 particles are used. We observe that particles initially spread across the space ($t = 0$). As the robot only receive partial information in the visual navigation task, particles gradually form a distribution with two clusters ($t = 56$), which represent two major hypotheses of its current state. After more information is incorporated into the belief, they begin to converge and finally become a single cluster ($t = 81$). We did not observe particle depletion and posterior collapse in our experiment. This could be better avoided by adding an entropy loss to the learned variance of $f_{\text{trans}}$ and we will leave it for future study.

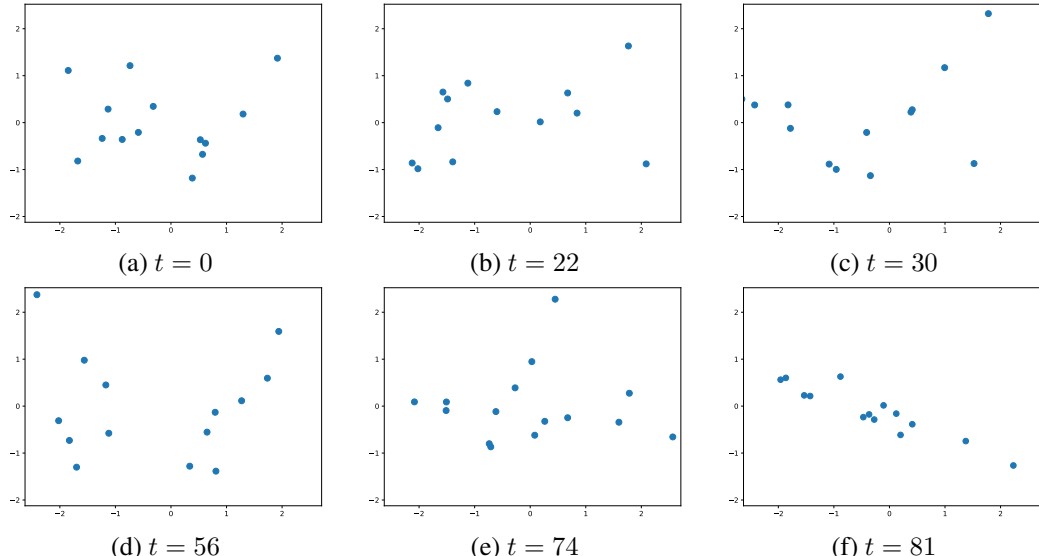

