# OpenReview forum: "Discriminative Particle Filter Reinforcement Learning for Complex Partial observations"
_ICLR.cc/2020/Conference — Accept (Poster)_

### Official Review · AnonReviewer1 · 2019-10-23
**Official Blind Review #1**

**Rating:** 8

**Review:**

Update: my concerns have been addressed and I have updated the score to 8
****

This paper introduces 3 neat ideas for training deep reinforcement learning (DRL) agents with state variables so that they can handle partially observed environments:
1) model the latent state variable as a belief distribution, using a collection of weighted hidden states, like in the particle filter (PF), with an explicit belief update of each particle, calculation of the weight using an observation function, and a differentiable re-weighting function to get the new belief distribution,
2) base the policy on the whole set of particles, by quantifying that set using its mean as well as a collection of K learnable moments (specifically, K Moment Generating Functions, each one corresponding to a dot product between the moment variable and the hidden state of the particle),
3) instead of generating the observations, take again the idea from PF which is to measure the agreement between the current observation o_t and the i-th particle state variable h_t^i, via a learnable discriminative function.
From what I understand, the only gradients in the model come from the usual 3 RL losses, and the observation functions in the discriminative PF are trained because they weigh the particles.

The model, trained using Advantage Actor Critic (A2C) works well on the (contrived, more on that later) "flickering Natural" Atari RL environment as well as on the Habitat navigation challenge, outperforming both the GRU-based deep RL agent and the Deep Variational RL based agent that uses variational sequence auto-encoders (and extra gradients from the observation function...). The ablation analysis confirms the advantages of the 3 ideas introduced in the paper.

The paper is a very well written and the experiments are very well executed. I believe that the idea is novel. I gave this paper only a weak accept because of unclear explanation and of several missed opportunities:

* The observation function f_{obs}(h_t^i, o_t) is insufficiently explained. I understood it was trained using discriminative training. Does it mean that different observations o_t are used, and if so, how many? Or is the observation o_t the current observation of the agent, but only the h_t^i change? In which case, what makes it discriminative? Isn't there a posterior collapse, with all particles ending up bearing the same state? Does the function f_{obs} input o_t or u(o_t), where u is the convolutional network?

* These questions could be easily answered with pseudocode in the appendix.

* In section 3.1, what is the relationship between p_t(i) and f_{obs}(h_t^i, o_t)?

* Particle filters in navigation enable to store the history of the observations of the mobile robot, accumulating the laser range scans and matching them to the observations. At the end, one can visualise the map stored in a given particle, as well as visualise the point cloud of the particle coordinates and show the trajectories of these particles. Here the particles contain the hidden states of the agent. Could you similarly to traditional PF, visualise the position of the agent by matching the point cloud {{h_t^i}_i}_t to a set of observations o_k taken from the whole environment, and plotting a 2D map of weights coming from function f_{obs}(h_t, o_k) evaluated over all k?

* In the discussion, can you comment on the relationship between Monte-Carlo Tree Search in RL agents (sampling different trajectories) vs. here (sampling different states)?

* While I understand the need to use that environment for the sake of comparison to DVRL, the Atari + flickering + natural images dataset is very artificial and contrived. I would be interested in seeing more analysis of the discriminative PF RL algorithm on navigation tasks, given that that's what PF were designed for.

Some missing references:
* Early references on DRL for navigation:
Zhu et al (2016) "Target-driven visual navigation in indoor scenes using deep reinforcement learning"
Lample & Chaplot (2016) "Playing FPS games with deep reinforcement learning"
Mirowski et al (2016) "Learning to navigate in complex environments"


**Experience Assessment:**

I have published in this field for several years.

**Review Assessment: Checking Correctness Of Derivations And Theory:**

I carefully checked the derivations and theory.

**Review Assessment: Checking Correctness Of Experiments:**

I assessed the sensibility of the experiments.

**Review Assessment: Thoroughness In Paper Reading:**

I read the paper thoroughly.

---

> ### Author Response · Authors · 2019-11-12
> **Response to Reviewer #1**
>
> Thank you for the positive feedback. We are grateful for the many suggestions for improvement, which we have mostly incorporated in the revised manuscript. We would like to further clarify some of the questions below.
>
> Q: The observation function f_{obs}(h_t^i, o_t) is insufficiently explained.
> A: Thank you for pointing this out. We have now revised the measurement update subsection in Sect. 3.1.
>
> Q: Does the function f_{obs} input o_t or u(o_t), where u is the convolutional network?
> A: The input to f_{obs} is o_t, while the input to f_{trans} is u(o_t). We use u(o_t) for the transition function instead of o_t to emphasize the connection to standard particle filtering, where the last control signal is usually an input to f_{trans}. In our case, this control signal is extracted from o_t, using u(o_t). This explains the dynamics of the environment, e.g., why the ghosts move in Atari games. We have added implementation details in Appendix B.1 and a detailed algorithm in Appendix D, which may clarify this question.
>
> Q: What is the relationship between p_t(i) and f_{obs}(h_t^i, o_t)?
> A: p_t(i) denotes the probability of sampling the particle i with the standard resampling step, and p_t(i) = w_t^i, the particle weight before resampling. f_{obs}(h_t^i, o_t) is the learned observation likelihood value. The relationship between w_t^i and f_{obs} is given by Eqn.(2), and p_t(i) = w_t^i. We have found a typo in Section 3.1 where f_{obs} was used instead of f_{trans} under the “Transition update” subsection. We believe this might have caused the confusion, which we have fixed now. Thank you for directing our attention to this issue.
>
> Q: In the navigation task, visualize the particles with respect to different observations
> A: Thank you for the suggestion. We recognize the importance of investigating what our model learned. To shed some light on this issue, we have visualized particles for the navigation task, and added it to Appendix E.3. Since the particles in DPFRL correspond to learn latent states, we applied a PCA decomposition and plotted the first two principal components through time. We can observe that particles are initially distributed across the space. With more observations, particles form two clusters, and finally, they converge around a single cluster.
>
> We would also like to mention that while evaluating the learned observation function f_obs would be also interesting, we cannot recover the generative model p(o|h) from f_obs. The outputs of f_obs(o | h) for a fixed h and different o inputs are not meaningfully comparable because f_obs is only trained to estimate unnormalized likelihood values for a fixed o, with different h_t inputs corresponding to particles at time t.
>
> Q: Isn't there a posterior collapse, with all particles ending up bearing the same state?
> A: Since particles are sampled from the stochastic transition function f_{trans} after each transition step, the particle distribution could indeed collapse to a single state if the variance of f_{trans} would be close to zero. We have not observed this happening in our experiments. This could also be prevented, e.g., by an additional entropy constraint on the f_{trans} output. An example of the evolution of particles is now added to Appendix E.3.
>
> Q: What’s the difference between the sampling the different trajectories in MCTS with the sampling states (PF) here?
> A: MCTS samples trajectories to construct a search tree, i.e., reason about the future to choose an action for the current time step. Particle filters sample states to approximate a belief distribution over the current partially observed state. In DPFRL we learn a model-free policy conditioned on the particle belief. Combining the two, particle filtering and forward search, would be an interesting direction for future work.
>
> Q: Natural Flickering Atari games are artificial and contrived.
> A: While we agree that the task is somewhat contrived, we believe it is valuable for demonstrating the issue of complex noisy observations. By adding background videos to Atari games we can control the observation complexity without affecting the difficulty of the decision making problem, enabling informative comparisons in a well-understood domain. We agree that eventually, interesting tasks will be more like the Habitat domain and not like Atari. We have updated the manuscript to better reflect our motivations for choosing domains.
>
> Q: Additional references and pseudocode
> A: Thank you for the useful suggestions. We have added the references to Section 4.2 and the pseudocode of our algorithm to Appendix D.

---

> > ### Comment · AnonReviewer1 · 2019-11-15
> > **Official Blind Review #1 update**
> >
> > Thank you for the clarifications, additional analyses and references. The paper is improved and I have updated the score accordingly.

---

> > > ### Author Response · Authors · 2019-11-15
> > > **We thank the reviewer for the encouragement**
> > >
> > > We thank the reviewer for the encouragement!
> > >
> > > We will further study the learned representation to provide intuitions for designing better representation learning algorithms.

---

### Official Review · AnonReviewer2 · 2019-10-25
**Official Blind Review #2**

**Rating:** 6

**Review:**

This is a well written paper. It introduces a principled method for POMDP RL:  Discriminative Particle Filter Reinforcement Learning (DPFRL).
It combines the strength of Bayesian filtering and policy-oriented discriminative modeling. DPFRL encodes a differentiable particle filter with learned transition & observation models in a neural network, allowing for reasoning with partial observations over multiple time steps. It performs explicit belief tracking with discriminative learnable particle filters optimized directly for the RL policy.

Experimental results show that DPFRL achieves state-of-the-art on POMDP RL benchmarks. I especially like the paper covers a diverse set of applications, including Mountain Hike, the classic Atari games, and visual navigation (Habitat). Improved performance is reported. Results show that the particle filter structure is effective for handling partial observations, and the discriminative parameterization allows for complex observations.


**Experience Assessment:**

I do not know much about this area.

**Review Assessment: Checking Correctness Of Derivations And Theory:**

I did not assess the derivations or theory.

**Review Assessment: Checking Correctness Of Experiments:**

I did not assess the experiments.

**Review Assessment: Thoroughness In Paper Reading:**

I read the paper at least twice and used my best judgement in assessing the paper.

---

> ### Author Response · Authors · 2019-11-12
> **Response to Reviewer #2**
>
> Thank you for the positive feedback. We have further revised the manuscript according to others suggestions.

---

### Official Review · AnonReviewer4 · 2019-11-01
**Official Blind Review #4**

**Rating:** 8

**Review:**

What is the specific question/problem tackled by the paper?

Representation learning in POMDPs in order to ignore spurious information in observations.


Is the approach well motivated, including being well-placed in the literature?

Some comparisons to related work are missing; while the comparisons would enrich the paper, their absence is not fundamentally limiting to the conclusions.

There's an additional PSR-related work that can be seen as learning representations for POMDPs (Guo et al., Neural predictive belief representations, arXiv:1811.06407). This work is in line with the work of Gregor et al., 2019, and both provide suitable representation learning techniques for POMDPs.

These representation learning in the paper is based on action-conditional predictions of future quantities, which is complementary to the approach proposed in the paper. That is, one could conceive adding action-conditional predictions of the future with the particles as the RNN states.


Does the paper support the claims? This includes determining if results, whether theoretical or empirical, are correct and if they are scientifically rigorous.

I think the support is somewhat adequate.

The claim that the proposed method handles spurious information is well supported by the experiment in mountain hike, but not quite so by the Atari experiments. The performance (upon introduction of the "natural" on top of flickering) takes a big hit for both DPFRL and DVRL. Still, the performance improvement of DPFRL over DVRL is still an encouraging result.


Summarize what the paper claims to do/contribute. Be positive and generous.

The paper proposes a neural implementation of particle filters, by treating samples of RNN states as particles. The particles are used to estimate moment-generating functions evaluated at trained vectors, which in turn are supposed to provide more information for the policy's decision making. The paper uses a discriminator to shape the representation.

The ablation study suggests that all three components (particles, MGFs & discrimination) are necessary. However, the third component has been shown not to be exclusively helpful for representation learning (Gregor et al., Guo et al.) I would suggest a study in comparison to Gregor et al.'s method (DRAW) instead.


Clearly state your decision (accept or reject) with one or two key reasons for this choice.

I vote for acceptance.


Provide supporting arguments for the reasons for the decision.

I think the algorithmic idea in this paper is a step in the right direction and can be of interest for the community. I would hope for the benchmarks to be more like the Habitat, and less like Atari with background videos. The conclusions in the latter benchmark seem less likely to apply to tasks in physically structured environments.


Provide additional feedback with the aim to improve the paper. Make it clear that these points are here to help, and not necessarily part of your decision assessment.

I think it is important for the paper to qualify the kind of POMDPs being considered. The defining features of most of the environments being used is that the state is observed through a noisy channel. Many POMDPs are of interest because the observations are really providing partial information about the state, even if it is noiseless. This is the case for the Habitat setting.

Because the paper's claims about the adequacy of the method for POMDPs rests on the choice of environments, I think it's important to quality what kind of POMDPs are being considered here. I would also caution against stating that the environment is closer to the real world. It would perhaps be better to say that the natural flickering is more interesting than the natural and the flickering because it benchmarks robustness to irrelevant information in observations, provided almost in tandem with state information; with intermittently missing observations.

Please add some explanation about how the negative examples are sampled for the contrastive estimation.

**Experience Assessment:**

I have published one or two papers in this area.

**Review Assessment: Checking Correctness Of Derivations And Theory:**

N/A

**Review Assessment: Checking Correctness Of Experiments:**

I assessed the sensibility of the experiments.

**Review Assessment: Thoroughness In Paper Reading:**

I read the paper at least twice and used my best judgement in assessing the paper.

---

> ### Author Response · Authors · 2019-11-12
> **Response to Reviewer #4**
>
> Thank you for your valuable feedback, and suggestions for improvement. We have revised the paper accordingly. We would also like to answer some of the questions below.
>
> Q: Additional references on PSR models
> A: Thank you for pointing us to the additional references. We have added them to the related work section.
>
> Q: How are the negative examples sampled for the contrastive estimation.
> A: Unlike contrastive estimation, DPFRL does not need explicit negative examples for training. At each step f_{obs} is evaluated for each particle of the belief that is updated with the particle filter algorithm. Specifically, particles are sampled from the learned stochastic transition function f_{trans}. Gradients for f_{obs} come from the RL loss, through the weighting of particles as shown in Eq.2. We have updated Section 3.1 to clarify how f_{obs} is trained.
>
> Q: Benefit of discriminative training & Additional comparison with DRAW
> A: Our ablation experiments with DPFRL-generative directly aim to compare DPFRL with discriminative vs. generative parameterization. We agree that more powerful generative models, such as DRAW, could potentially improve the performance of our generative baselines. While comparing with stronger generative methods would be interesting, their computational cost is prohibitive in our case because we need to evaluate the model for each of the K particles. We have added a discussion in the related work section in the updated manuscript.
>
> Q: Adding the PSR-like action conditioned prediction structure to DPFRL
> A: Yes, adding the action conditioned prediction structure would benefit regularizing the latent representations, especially when learning a better latent dynamics. Currently, we mainly focus on how to structure the latent belief (using latent particle filter) and how to condition the policy on the whole belief. Introducing a PSR-like structure into DPFRL would be an interesting direction for future work.
>
> Q: Natural flickering Atari games are artificial & types of POMDPs considered in the paper.
> A: Indeed, we see the benefit of the natural flickering Atari in that they provide a controlled benchmark to understand the influence of complex observations in POMDPs. We agree that flickering observations do not capture the most interesting class of POMDPs and that Natural flickering Atari games are still far from real-world applications. We have chosen Mountain Hike and flickering Ataris games mainly for demonstration purposes in well-understood settings, and the Habitat domain for a more realistic application with challenging partial observability. We have updated the section 4.3 to better reflect these motivations.

---

### Decision · Program_Chairs · 2019-12-19

**Decision:**

Accept (Poster)

**Comment:**

The authors introduce an RL algorithm / architecture for partially observable
environments.
At the heart of it is a filtering algorithm based on a differentiable version of
sequential Monte Carlo inference.
The inferred particles are fed into a policy head and the whole architecture is
trained by RL.
The proposed methods was evaluated on multiple environments and ablations
establish that all moving parts are necessary for the observed performance.

All reviewers agree that this is an interesting contribution for addressing the
important problem of acting in POMDPs.

I think this paper is well above acceptance threshold. However, I have a few points that I
would quibble with:
1) I don't see how the proposed trampling is fully differentiable; as far as I
understand it, no credit is assigned to the discrete decision which particle to
reuse. Adding a uniform component to the resampling distribution does not
make it fully differentiable, see eg [Filtering Variational Objectives. Maddison
et al]. I think the authors might use a form of straight-through gradient approximation.
2) Just stating that unsupervised losses might incentivise the filter to learn
the wrong things, and just going back to plain RL loss is not in itself a novel
contribution; in extremely sparse reward settings, this will not be
satisfactory.